# Soil networks become more connected and take up more carbon as nature restoration progresses

Elly Morriën[1,2,*], S Emilia Hannula[3,*], L. Basten Snoek[1,4], Nico R. Helmsing[5], Hans Zweers[3], Mattias de Hollander[3], Raquel Luján Soto[1], Marie-Lara Bouffaud[6], Marc Buée[7,8], Wim Dimmers[9], Henk Duyts[1], Stefan Geisen[1,10], Mariangela Girlanda[11,12], Rob I. Griffiths[13], Helene-Bracht Jørgensen[14], John Jensen[15], Pierre Plassart[6], Dirk Redecker[16], Rűdiger M. Schmelz[17,18], Olaf Schmidt[19,20], Bruce C. Thomson[13], Emilie Tisserant[7,8], Stephane Uroz[7,8], Anne Winding[21], Mark J. Bailey[13], Michael Bonkowski[10], Jack H. Faber[9], Francis Martin[7,8], Philippe Lemanceau[6], Wietse de Boer[3,22], Johannes A. van Veen[3,23] & Wim H. van der Putten[1,4]

Soil organisms have an important role in aboveground community dynamics and ecosystem functioning in terrestrial ecosystems. However, most studies have considered soil biota as a black box or focussed on specific groups, whereas little is known about entire soil networks. Here we show that during the course of nature restoration on abandoned arable land a compositional shift in soil biota, preceded by tightening of the belowground networks, corresponds with enhanced efficiency of carbon uptake. In mid- and long-term abandoned field soil, carbon uptake by fungi increases without an increase in fungal biomass or shift in bacterial-to-fungal ratio. The implication of our findings is that during nature restoration the efficiency of nutrient cycling and carbon uptake can increase by a shift in fungal composition and/or fungal activity. Therefore, we propose that relationships between soil food web structure and carbon cycling in soils need to be reconsidered.

---

[1] NIOO-KNAW, Terrestrial Ecology, Droevendaalsesteeg 10, Wageningen 6708 PB, The Netherlands. [2] Institute for Biodiversity and Ecosystem Dynamics, Earth Surface Sciences Group (IBED-ESS), University of Amsterdam, PO Box 94246, Amsterdam 1090 GE, The Netherlands. [3] NIOO-KNAW, Microbial Ecology, Droevendaalsesteeg 10, Wageningen 6708 PB, The Netherlands. [4] Laboratory of Nematology, Wageningen University, Droevendaalsesteeg 1, Wageningen 6708 PB, The Netherlands. [5] NIOO-KNAW, Aquatic Ecology, Droevendaalsesteeg 10, Wageningen 6708 PB, The Netherlands. [6] Agroécologie, AgroSup Dijon, INRA, Univ. Bourgogne Franche-Comté, Dijon F-21000, France. [7] INRA, UMR 1136 'Interactions Arbres Micro-organisms', Centre INRA de Nancy, Champenoux F-54280, France. [8] Université de Lorraine, UMR 1136 'Interactions Arbres Micro-organisms' Vandoeuvre-les-Nancy F-54000, France. [9] Wageningen Environmental Research, Droevendaalsesteeg 3, P.O. Box 47, Wageningen 6700 AA, The Netherlands. [10] Department of Terrestrial Ecology, Institute of Zoology, University of Cologne, Zülpicher Str 47b, Cologne 50674, Germany. [11] Department Scienze della Vita e Biologia dei Sistemi, University of Torino, Viale Mattioli 25, Torino 10125, Italy. [12] National Research Council, Istituto per la Protezione Sostenibile delle Piante (IPSP-CNR), Viale Mattioli 25, 10125 Torino, Italy. [13] NERC Centre for Ecology & Hydrology, Benson Lane, Growmarch Gifford, Wallingford, Oxfordshire, Oxford 108, UK. [14] Department of Biology, Lund University, Lund SE-22362, Sweden. [15] Department of Bioscience, Aarhus University, Vejlsøvej 25, Silkeborg 8600, Denmark. [16] Agroécologie, AgroSup Dijon, INRA, Univ. Bourgogne Franche0Comté, Dijon Cedex F-21065, France. [17] ECT Oekotoxikologie GmbH, Böttgerstr. 2-14, Flörsheim 65439, Germany. [18] Department of Animal Biology, Plant Biology and Ecology, Science Faculty, Universidad de A Coruña, Rua da Fraga 1, Coruña 15008 A, Spain. [19] UCD School of Agriculture and Food Science, University College Dublin, Dublin 4, Ireland. [20] UCD Earth Institute, University College Dublin, Dublin 4, Ireland. [21] Department of Environmental Science, Aarhus University, Frederiksborgvej 399, PO Box 358, Roskilde 4000, Denmark. [22] Department of Soil Quality, Wageningen University, PO Box 47, Wageningen 6700AA, The Netherlands. [23] Department of Plant Ecology and Phytochemistry, Leiden University, PO Box 9505, Leiden 2300 RA, The Netherlands. * These authors contributed equally to this work. Correspondence and requests for materials should be addressed to E.M. (email: e.morrien@nioo.knaw.nl).

Many ecosystems worldwide face exposure to intensified human use[1-3], which has resulted in loss of biodiversity[4], altered functioning[5] and altered provisioning of ecosystem services[6]. The abandonment of disturbed land represents one of the most widely used restoration strategies implemented at a global scale[7], with the potential to promote biodiversity, and associated ecosystem services. However, the restoration of natural ecosystem functioning and soil properties is known to be a long-term process[7,8], dependent upon the time it takes to restore connections between different components of the community[9]. Over half a century ago, Odum[10] identified mechanistic linkages between the successional dynamics of natural communities and the functioning of natural ecosystems. Specifically, as communities progress through succession, diversity is expected to increase and nutrients will become 'locked-up' in the biota, with consequences for the build-up of soil organic matter and closure of the mineral cycles[10]. More recently, the interplay between aboveground and belowground biodiversity has emerged as a prominent determinant of the successional dynamics in biological communities[11]. However, little is known about how changes in the soil biota contribute to the associated changes in ecosystem functioning.

In ecosystems undergoing secondary succession, it is evident that available nitrogen diminishes, primary productivity decreases and the plant community shifts from fast- to slow-growing plant species[12]. There is less evidence of an increase of soil biodiversity[13], and evidence of a relationship between soil biodiversity and ecosystem functioning is mixed, at best[5,14-16]. As a result, it is still unclear how soil and plant community composition relate to each other and what is the relative role of plants and soil biota in driving soil processes and plant community development[12,17].

Interestingly, studies on a time series (chronosequence) of abandoned arable fields revealed that carbon and nitrogen mineralization by the soil food web increases during secondary succession[18]. This implies a more active soil microbial community in later successional stages[19-21] where bacterial-dominated systems are expected to be replaced by fungal-dominated systems[22] with more carbon turnover via fungi[23] and their consumers[24]. However, data to test these assumptions are largely lacking. Therefore, the aim of the present study was to examine how biodiversity, composition and structure of the soil community change during successional development of restored ecosystems.

We used a well-established chronosequence of nature restoration sites on ex-arable, formerly cultivated, lands that represent over 30 years of nature restoration. We determined biodiversity of almost all taxonomic groups of soil biota, analysed their network structure and added labelled carbon dioxide and mineral nitrogen to intact plant–soil systems in order to track their uptake by the soil food web. We tested the hypothesis that functional changes in carbon and nitrogen flows relate more strongly to the belowground community network structure than to belowground biodiversity.

We analysed variations in species co-occurrence and considered enhanced correlations as network tightening, which we define as a 'significant increase in percentage connectance and an increase in the strong correlations as a percentage of all possible correlations'[25]. Our results reveal increased tightening and, therefore, connectance, of the belowground networks during nature restoration on the ex-arable land. A combination of correlation-based network analysis and isotope labelling shows that soil network tightening corresponds with enhanced efficiency of the carbon uptake in the fungal channel of the soil food web, without an increase in the total amount of soil biodiversity or in fungal-to-bacterial biomass ratios. For nitrogen, the non-microbial species groups revealed a similar pattern as for carbon. Tightening of the networks reflects stronger co-occurring patterns of variation in soil biota[25]. Increased carbon and nitrogen uptake capacity by the fungal channel in the soil food web can be explained by stronger co-occurrence of preys and their predators[24], which enhances the efficiency of resource transfer in the soil food web compared with a soil food web where preys and predators are spatially isolated.

## Results

**Network structure**. During the course of succession following land abandonment, there was an increase in the number of strong correlations between groups of soil organisms based on species abundance data with Spearman's rank correlation >0.9 (Fig. 1a, Table 1). Network structure change was the most pronounced between recently and mid-term abandoned fields, largely owing to increased correlations between bacteria and fungi (Fig. 1b, Table 1). Analysis of co-occurrence showed that patterns in network structure were robust for the type of comparison; network analysis using presence–absence data in the correlation matrix showed the same transition in network tightening between recent and mid-term abandonment stages (Supplementary Figs 1 and 2).

During succession, the numbers of plant species declined (Supplementary Fig. 3 and Supplementary Table 1, respectively), plant species composition changed and plant community structure became less even, as is indicated by reduced $H$-value in the longer-term abandoned fields (Fig. 2, Table 2). Variation in abiotic soil properties was significantly higher in the recently abandoned fields than in the mid-term abandoned fields; however, there was no significant difference between variation in recent versus long-term abandoned fields (Supplementary Fig. 4). Abiotic conditions explained a substantial amount of variation of the different groups of soil biota (Supplementary Table 2). However, the increased network tightening from recent to long-term abandoned fields could not be explained by significantly declined variation in abiotic conditions.

The number of taxa in bacteria and most fungi showed a hump-shaped pattern, whereas numbers of taxa of arbuscular mycorrhizal fungi (AMF) significantly increased with progressing succession (Supplementary Fig. 3, Supplementary Table 1). The number of taxa of fungivorous cryptostigmatic mites, predaceous mesostigmatic mites, root-feeding nematodes and bacterivorous nematodes in general also increased during the course of succession, whereas other species groups did not show any successional change at all (Supplementary Fig. 3, Supplementary Table 1). On the other hand, there were significant changes in soil community composition, among others, in composition of bacteria, fungi and their predators (Table 2). Therefore, increased network tightening could not be explained only by a general convergence in plant community composition or soil properties or by the total amount of soil biodiversity, whereas a contribution of changed composition of the soil community could not be excluded.

**Stable isotope data**. Analysis of [13]C revealed that the tightening of the belowground networks coincided with increased efficiency of carbon uptake: in later successional stages that had been abandoned longer time ago, plants tended to have least newly photosynthesized carbon in their roots, whereas consumers, such as root-feeding nematodes and soil fungi, contained most of the supplied label (Fig. 3). This pattern becomes even clearer when considering the relative amounts of carbon in the microbes 1 day after pulse labelling (phospholipid fatty acids (PLFA): bacteria $F_{2,13} = 6.51$, $P = 0.01$, fungi $F_{2,13} = 2.85$,

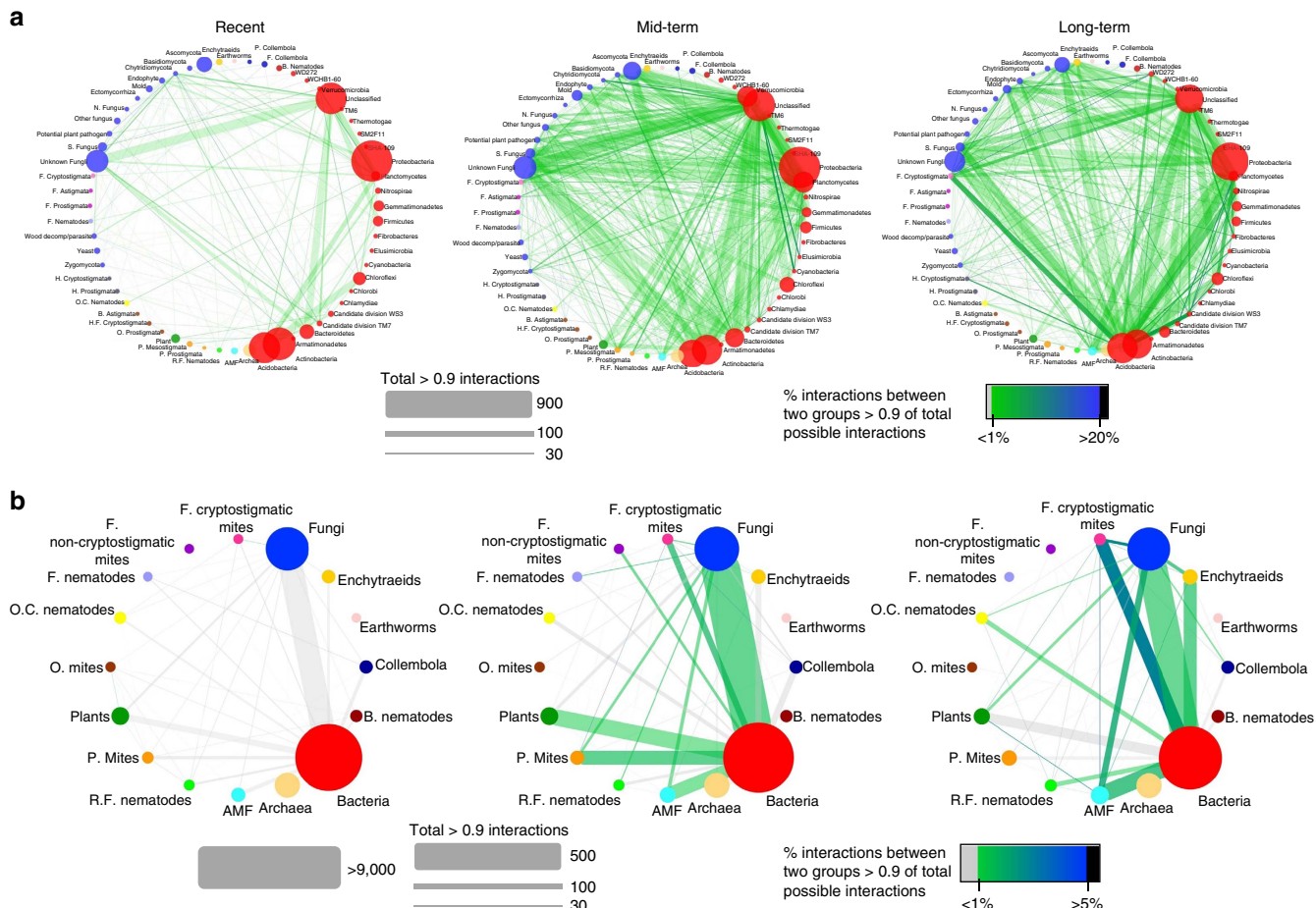

**Figure 1 | Network visualization of the interaction strengths.** Interaction strength between the species subgroups (**a**) and main species groups (**b**) in seminatural grasslands on recently, mid-term and long-term abandoned agricultural fields. Spearman's rank correlations of the relative abundances of all individual species combinations between two groups where calculated. The proportion of correlations >0.9 was divided by the total number of possible interactions to obtain the interaction strength between two groups of species. Line width is proportional to the absolute number of correlations >0.9. Line colour and transparency is proportional to the interaction strength, as indicated in the legend in the figure. The size of the circles is proportional to the number of species/taxa in that group. Red-filled circles are bacterial groups, blue-filled circles are fungal groups. Filled circles of other colours represent other taxa, with identities shown on the figure. B, bacterivorous; F, fungivorous; H, herbivorous; H.F, herbofungivorous; N, nematophagous; O, omnivorous; O.C., omni-carnivorous; P, predaceous; R.F., root-feeding; S., saprotrophic.

**Table 1 | Connectance calculated for the networks.**

| Subgroups | Recent | Mid-term | Long-term | Main groups | Recent | Mid-term | Long-term |
|---|---|---|---|---|---|---|---|
| Correlations >0.9 | 10,961 | 26,571 | 19,308 | Correlations >0.9 | 4,833 | 12,621 | 9,029 |
| All possible correlations | 1,749,816 | 2,239,795 | 1,510,742 | All possible correlations | 822,361 | 1,057,646 | 786,379 |
| Connectance % | 0.626 | 1.186 | 1.278 | Connectance % | 0.588 | 1.193 | 1.148 |

Connectance calculated for the networks in Fig. 1. For recent, mid-term and long-term abandonment, all correlations >0.9 (represented in Fig. 1) divided by all possible connections between the members of the nodes.

$P = 0.09$, neutral lipid fatty acids (NLFA): AMF $F_{2,13} = 1.16$, $P = 0.34$) and, later, in consumers and their predators (Fig. 4). In the recently abandoned grasslands, fungi took up half of the carbon, whereas in long-term abandoned grasslands three quarters of the carbon was taken up by fungi. These changes could not be explained by increased fungal biomass or by an increase in fungal-to-bacterial biomass ratio (Figs 3 and 5, respectively). The changes, however, go along with substantial shifts in microbial consumers. The combination of tighter connections and stronger labelling of the fungal channel in the mid and longer-term abandoned fields make us conclude

that network tightening contributes to enhanced efficiency of carbon uptake by the soil food web.

In early successional stages at recently abandoned fields, fungivorous collembola and nematodes were the predominant fungal consumers, whereas in later succession stages mites took a larger proportion of the labelled carbon (Fig. 4). Interestingly, these differences in soil community functioning were recorded in spite of soil cores being collected from sites along the chronosequence that were largely dominated by the same three plant species (Supplementary Fig. 5). Therefore, our results suggest that successional changes in soil community

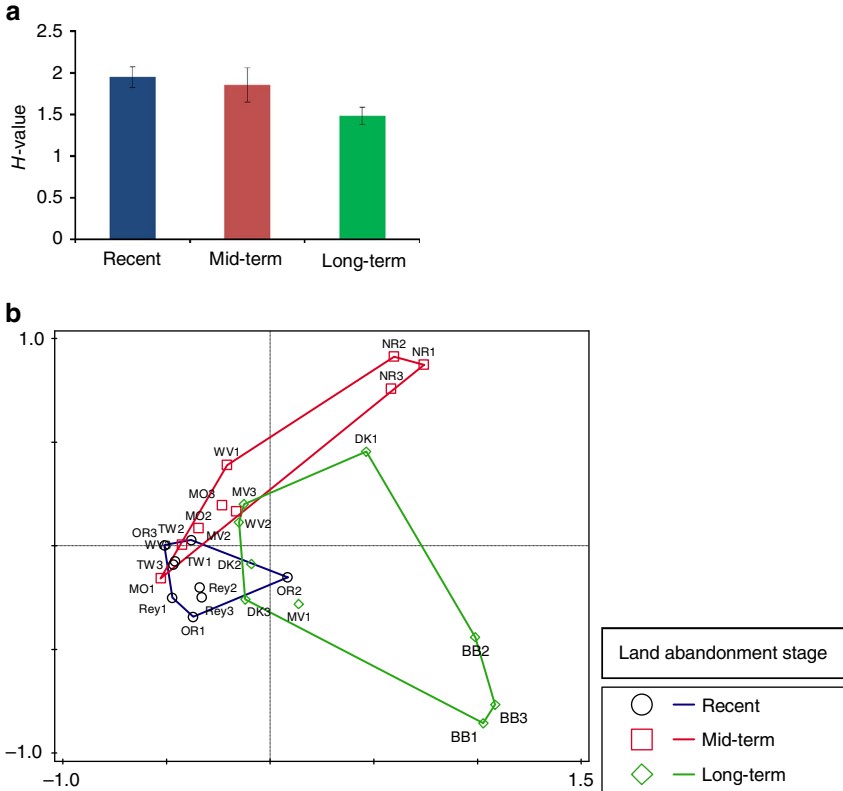

**Figure 2 | Plant species assemblage.** (**a**) Average $H$-values of recent, mid-term and long-term plant communities. (**b**) Principal coordinate analysis (PCO) on presence–absence data of the plant species in the field sites. Statistical summary on the difference between recent, mid-term and long-term sites is presented in Table 2 under analysis of similarities of the plant community in the field sites where the experimental cores were extracted.

**Table 2 | PERMANOVA and ANOSIM results on changes in community composition.**

| | PERMANOVA on abundance data | | | | Significant difference between groups | | | ANOSIM on presence–absence data | | | | Significant difference between groups | | |
|---|---|---|---|---|---|---|---|---|---|---|---|---|---|---|
| | Total SS | Within group SS | F | P | Recent-mid-term | Recent-long-term | Mid-term-long-term | Mean rank within | Mean rank between | R | P | Recent-mid-term | Recent-long-term | Mid-term-long-term |
| Plants | 4.71E + 04 | 3.65E + 04 | 3.486 | **0.0015** | No | Yes | Yes | 146.5 | 189.1 | 0.2425 | **0.0009** | Yes | Yes | No |
| Archeaea (TRFLP data) | 2.413 | 1.765 | 3.855 | **0.0084** | Yes | Yes | No | | | | | | | |
| Bacteria | 6.98E + 05 | 5.79E + 05 | 2.365 | **0.0063** | Yes | No | No | 123.6 | 180.5 | 0.3499 | **0.0001** | Yes | No | Yes |
| Fungi | 1.34E + 05 | 1.10E + 05 | 2.581 | **0.0001** | Yes | Yes | Yes | 116.3 | 183.8 | 0.4156 | **0.0001** | Yes | Yes | Yes |
| Protists | 2.79E + 08 | 2.27E + 08 | 0.8014 | 0.6118 | No | No | No | | | | | | | |
| Nematodes | 2.32E + 08 | 2.19E + 08 | 0.7375 | 0.7857 | No | No | No | 145.2 | 189.7 | 0.2532 | **0.0009** | No | Yes | No |
| Enchytraeids | 2.34E + 05 | 2.11E + 05 | 1.303 | 0.1336 | No | Yes | No | 150.8 | 187.2 | 0.2076 | **0.0004** | Yes | Yes | Yes |
| Collembola | 5.49E + 04 | 4.93E + 04 | 1.377 | 0.2277 | No | No | No | 174.4 | 176.7 | 0.01288 | 0.3532 | No | No | No |
| Mites | 7.55E + 10 | 7.04E + 10 | 0.881 | 0.5871 | No | No | No | 141.3 | 191.4 | 0.2855 | **0.0005** | Yes | Yes | No |
| Earthworms | 170.6 | 153.6 | 1.331 | 0.2673 | No | No | No | | | | | | | |

ANOSIM, analysis of similarities; PERMANOVA, permutational multivariate analysis of variance; TRFLP, terminal restriction length polymorphism.
PERMANOVA results on changes in community composition of plants, archaea, bacteria, fungi, protists, nematodes, enchytraeids, collembolan, mites and earthworms. In case of clear differences between abundance data and presence–absence data, an additional ANOSIM analysis was performed. Significant $P$ values are marked in bold. Most groups did change in community assemblage over successional stage.

structure and functioning can arise even under the same plant community composition. Such field-based evidence on the role of whole-soil biodiversity in ecosystem functioning is quite rare[2,16]. Detailed analysis of incorporation of label into the soil food web revealed similar temporal patterns of incorporation of $^{13}C$ and $^{15}N$ into higher trophic levels. It is possible to analyse $^{15}N$ in microbes, but methods do not allow distinguishing bacterial from fungal $^{15}N$. Therefore, we chose not to relate tightening of the belowground networks to the

microbial efficiency of nitrogen use by the belowground food web (Supplementary Tables 3 and 4 and Supplementary Fig. 6).

## Discussion

We show that nature restoration on ex-arable land results in increased connectance of the soil biota, which leads to increased tightening of the networks of soil biota. Increased network tightening may be due to several factors. First, tightening

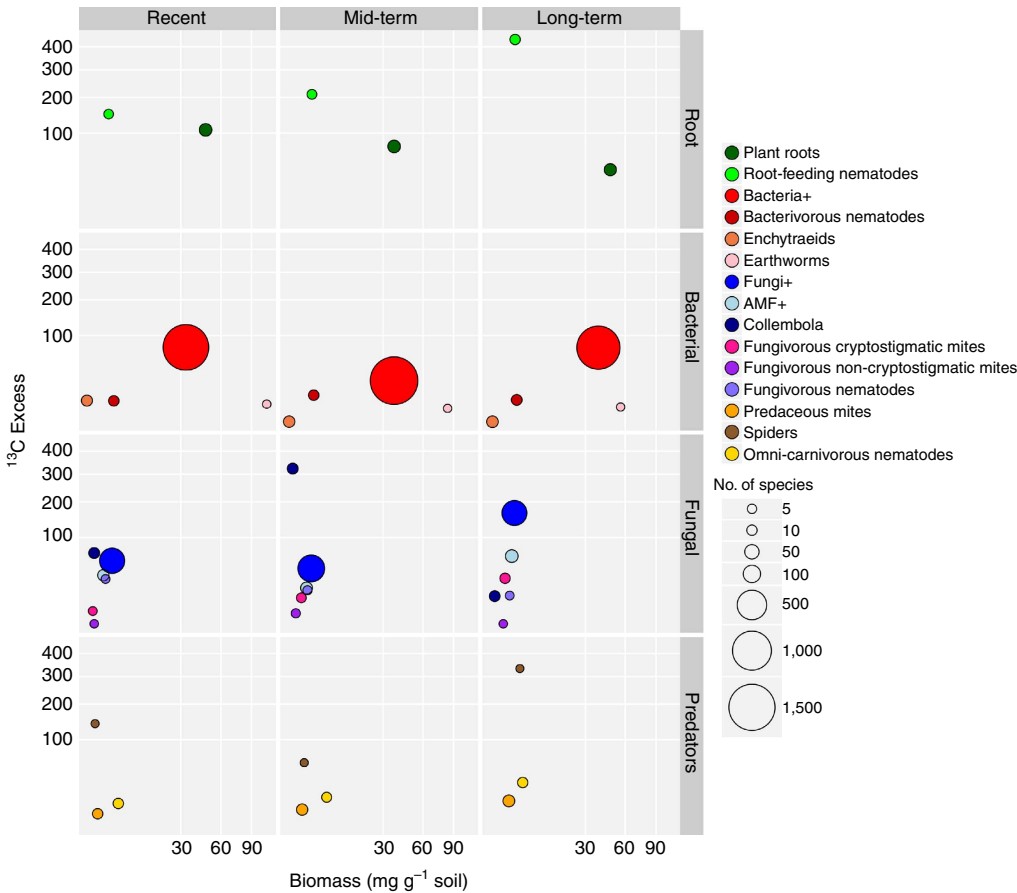

**Figure 3 | Carbon flow in relation to biomass and abundance in the soil food web.** Labelled carbon derived from living components in the soil: roots (green), bacterial channel (red, orange and pink), fungal channel (blue, purple, magenta), and higher trophic levels (brown, yellow, orange). The groups indicated with $+$ represent the amount of $^{13}C$ excess in pmol per gram soil (bacteria, fungi, AMF) measured 1 day after pulse labelling. For all other groups, the $^{13}C$ excess is the increase in $\delta^{13}C$ values of the labelled compared with natural values, measured from non-labelled controls, in recently, mid-term and long-term abandoned agricultural fields. Labelled compounds in plant roots have been measured 1 day after pulse labelling. Labels in root-feeding nematodes, bacterivorous nematodes, enchytraeids, earthworms, collembolans, fungivorous cryptostigmatic mites and fungivorous nematodes have been determined 1 week after pulse labelling, and fungivorous non-cryptostigmatic mites, predaceous mites, spiders and omni-carnivorous nematodes were determined 2 weeks after pulse labelling.

may be caused by successional shifts in species[26]. Bacteria and fungi showed hump-shaped development in numbers of taxa, whereas numbers of AMF taxa steadily increased, indirectly suggesting that there are indeed shifts in species composition along the successional gradient. AMF have been suggested to increasingly influence plant community composition with increasing time since land abandonment[27]. However, in our study network tightening is due to changes in more species groups than AMF alone. Second, increased tightening could be due to declined nutrient availability in the soil along the successional gradient[18,28,29], which may enhance the necessity of stronger cooperative and trophic interactions between functional groups of soil biota.

Third, changes in the soil physical conditions can influence network tightening[30]. Arable soils are assumed to be relatively heterogeneous[31,32], whereas natural succession following land abandonment will increase spatial heterogeneity in abiotic soil conditions[33]. Soil biota have a variety of responses to soil heterogeneity[34]. Increased soil heterogeneity could contribute to network tightening, when it enhances co-occurrence patterns of variation in the soil biota. We found reduced variation in soil abiotic properties from recent to mid-term abandoned fields, but there were no differences in variation between recent and

longer-term abandonment stages, which only partly supports the possibility that changes in soil abiotic factors enhance network tightening. Further correlative analyses of soil abiotic properties and network tightening would require independent pairs, however, we do not have individual networks for each individual soil sample used for abiotic analyses.

Our $^{13}C/^{15}N$ analyses revealed that a plant community dominated by the same species allocated less carbon and nitrogen to the roots in soil with late (long-term abandoned) than in soil with early successional (recently abandoned) soil communities but that the mid-late successional soil communities were more efficient in carbon uptake. It may be that low abundant plant species[35] or conversion of soil abiotic properties have changed soil functioning, but our results also support the suggestion that changes in soil community structure may precede succession in plant communities[16,17].

Opposite to expected, during successional transition the fungal biomass and the fungal-to-bacterial biomass ratios did not increase. Thus nature restoration resulted in a transition in terms of belowground taxonomical composition and fungal productivity but not in terms of fungal biomass. Interestingly, saprotrophic fungi represented only 0.06–0.08 of the fungal-to-bacterial ratio of the total microbial biomass in PLFAs, which

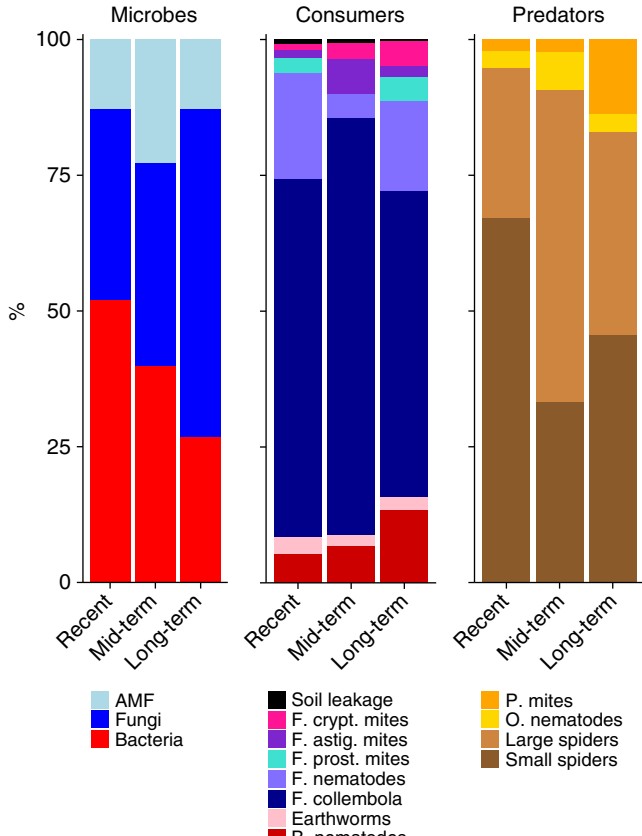

**Figure 4 | Relative carbon amounts in trophic level along abandonment stage.** The carbon measured at the relevant time points for each group of soil biota. At day 1, the carbon is distributed among microbes. The microbe panel represents relative amounts of carbon exuded by the roots at day 1 after labelling. The total amount of labelled carbon in the roots decreases during succession (Supplementary Fig. 6). We have therefore presented the relative distribution of carbon scaled to the total amount of labelled carbon in the roots as excess $^{13}C$ (the increase in atom% C values of the labelled compared with natural values measured from non-labelled controls) (bacteria, fungi, AMF). Bacteria (red), fungi (blue) and AMF (light blue) receive carbon from the plant roots. This carbon is distributed into the fungal channel and bacterial channel, where 1 week after labelling it is taken up by fungivorous mites, nematodes, collembola and bacterivorous nematodes and earthworms scaled to the total amount of labelled carbon in the roots as excess $^{13}C$. After 2 weeks after labeling, the carbon had reached the predators: spiders (brown), predaceous mites (orange), and omnivorous nematodes (yellow). Values of labels in the predators were also scaled to the total amount of labelled carbon in the roots as excess $^{13}C$. Absolute values for these groups are shown in Fig. 2. B, bacterivorous; F, fungivorous; O, omnivorous; P, predaceous.

is in accordance with previous estimates[36], yet these fungi processed most of the carbon in later successional stages (Fig. 4)[23]. Such changes in soil community structure and functioning have been rarely considered in relation to nature restoration[9]. Often, restoration targets are focussing on aboveground biodiversity and the presence of rare or red list species, although it has been demonstrated that adding particular soil inocula can direct vegetation development towards particular target systems[37].

We conclude that over successional time the connectance of species in the soil community increases, while carbon uptake becomes more efficient, even without major changes in species composition of the dominant plants. Our network approach combined with labelling study concerns a substantially different approach compared with previous soil food web modelling studies[18,38], because it is based on actual community composition, whereas food web models are based on biomass of entire feeding groups. Our results suggest that transition in fungal composition can change element cycling and carbon uptake in soil without an increase in fungal biomass or fungal-to-bacterial biomass ratio. We propose that there is a need to verify these findings also in other chronosequences and re-think how soil food web structure influences carbon cycling in soils.

## Methods

**Ex-arable land chronosequence.** We used a well-established chronosequence[21,39–41] of nine ex-arable fields all on Pleistocene sandy soils. The history of agricultural use is comparable; on all fields, there was a crop rotation, including barley and potato. The fields were abandoned from agricultural practice at different points in time. Following abandonment, seminatural grasslands were allowed to establish, all fields were grazed by free-ranging cattle and additionally mowed maximally once per year. On 18 and 19 October 2011, we visited the field sites marked in Supplementary Fig. 7 that correspond with the coordinates provided in Supplementary Table 5. At each ex-arable field, we collected soil and plant samples from three subplots of two square metres each, which were 100 metres apart from each other. In one square metre, vegetation records were made, whereas in the other square metre soil cores were collected from the top 10 cm for analysing microorganism composition and soil properties and for collecting enchytraeids, nematodes and soil micro-fauna by extraction methods. The soil samples were collected using a split-corer sampling device. In the same square, we collected earthworms by a combination of hand-sorting of $30 \times 30 \times 30$ cm$_3$ soil monoliths excavated with a hand-held spade and subsequent liquid irritant extraction of earthworms from the deeper soil layer. Aboveground and belowground standing plant biomass were determined based on these same excavated soil monoliths. Soil samples for microbial identification were processed the day after collection and transported by courier to specialists in our research team for further identification and quantification. Samples for soil analyses, nematodes and enchytraeids were stored and transported at 4 °C until processing. Soil micro-fauna core rings were processed the same day.

**Plants.** For the vegetation records in the square metres, first the percentage of bare soil, forbs and mosses was estimated and then the percentage cover of all plant species present. The estimates per plant species as percentage cover were used in the network analysis as a measure of plant abundance. Biomasses and C/N ratio of the plant material in the cores are presented in Supplementary Fig. 8.

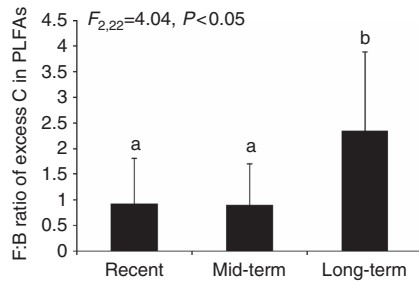
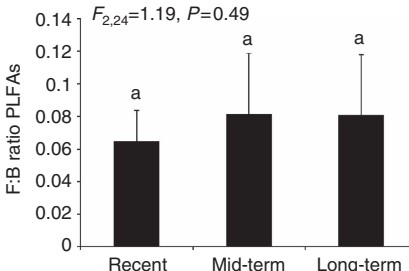

**Figure 5 | Fungal-to-bacterial ratios in the three abandonment stages.** Left panel: fungal (F) to bacterial (B) ratio of excess C (labelled excess in contrast to unlabelled controls) in PLFA. Right panel: the total F–B ratios in PLFAs. Error bars represent s.d.

**Microorganisms.** The soil samples collected for isolating DNA of microorganisms were sieved using a 5-mm mesh size to remove stones and roots. Sieved samples were transferred to INRA Dijon (France) for DNA extraction following a Standard Operating Procedure[42], where after the DNA extracts were distributed to the various co-workers for further analyses (see Supplementary Methods for details on sequencing). Separate samples were prepared for extraction of protists and sent to the University of Cologne (Germany).

**Archaea.** For archaeal communities a simple community profiling technique (terminal restriction length polymorphism) was used according to the methods utilized by Thomson *et al.*[43]. Archeal DNA was amplified using primers A364aF (fluorescently labelled) and A934b[44]. Amplicons were then digested using TaqI. Fragment analysis was subsequently carried out using a 3730 DNA analyser (Life Technologies, Paisley, UK).

**Protists.** Protists were extracted and enumerated simultaneously at the University of Cologne (Germany) using a modified Liquid Aliquot Method[45]. Protists were morphologically identified up to genus level using an inverted microscope (Nikon Eclipse TS100, Japan) at × 400 magnification. The abundance data obtained using this method were used in the network analysis. As protists were not counted in all replicates, they were excluded from the network analyses.

**Micro-fauna.** Micro-fauna was extracted from the split-core rings using a Tullgren extraction setting for 7 days at Wageningen Environmental Research (The Netherlands) following ISO standards[46]. We followed a two-step extraction, with a 3 days' initial temperature of 28 °C and a subsequent 4-day period at 45 °C, using a heat-generating carbon wire light bulb of 60 Watt above the samples. Collembolans were identified visually using a reversed light microscope at Aarhus University (Denmark) while mites were visually identified using a microscope at Wageningen Environmental Research.

**Nematodes.** Nematodes were extracted from 100 ml of soil using Oostenbrink elutriators[47]. Roots occurring in the sample were used to collect root-inhabiting nematodes (see below). The suspensions with nematodes were led through one 75-μm sieve and three 45-μm sieves. The material, including nematodes collected from the 75- and 45-μm sieves, was transferred to a double cotton filter (Hygia rapid, Hartmann AG, Heidenheim, Germany) on a sieve in a dish with a layer of tap water[47]. The nematodes were allowed to migrate through the filter into the water for 48 h at room temperature, which resulted in relatively clean suspensions for nematode counting. Suspensions were stored at 4 °C until they were fixated with hot and then cold 4% formalin. Root-inhabiting nematodes were collected using a mistifier. After nematode extraction for 48 h in the mistifier, the roots were air-dried and weighted. The total numbers of nematodes in the root were counted and standardized to dry root weight of extracted material; for soil, the samples were always extracted from 100 ml volume of fresh soil. They were identified to genus or family level using a reverse light microscope and categorized into feeding guilds according to (ref. 48) and (ref. 49). Abundances were used in the network analysis.

**Enchytraeids.** Enchytraeids were extracted from the soil cores with the hot/wet funnel method by O'Connor[50] following ISO standards[51]. Specimen were identified to species *in vivo* with a stereolupe (× 10– × 40 magnification) and a light microscope equipped with interference contrast (Nomarksi) optics (× 40– × 400 magnification), using the keys and techniques in Schmelz and Rut[52,53], together with primary literature. Most of the specimen (>95%) were identified to species level; the remainder was identified to genus level and abundances assorted proportionally to the species found in the sample. A reference collection of species was established with specimens fixed and stored appropriately for sequencing (DNA barcodes) and morphological reinvestigation.

**Earthworms.** Earthworms were extracted by a combination of active (soil hand-sorting) and passive allyl isothiocyanate (AITC) extraction methods. First, we hand-sorted earthworms from the $30 \times 30 \times 30$ cm$^3$ soil monoliths that were taken after clipping the aboveground biomass and before the soil was sieved to determine the standing root biomass (not shown). Then a weak mustard oil (100 mg AITC per litre) solution was poured into each pit (repeated once after about 10 min, totalling 10 l), and emerging earthworms were removed, rinsed in tap water and added to hand-extracted worms. The soil monoliths were stored at 4 °C and processed under laboratory conditions 2 days after collection. The collected earthworms were rinsed, weighed alive (with gut content), fixed in 4% formalin and, after a week, transferred to 70% ethanol. Adult and subadult individuals with sufficient sexual features were identified to species level based on external morphology, using Sims and Gerard[54]. Juveniles were identified to genera.

**Network analyses.** We removed single-sample occurrences per land abandonment stage before creating the Spearman's rank correlation matrix based on abundance data for preparation of the visualization of the correlation matrix using Cytoscape[55]. We used aggregated groups consisting of species that are known to share a common function (that is, AMF). If function was unknown (that is, for bacteria and archaea), taxonomical classification was used (Supplementary Tables 5 and 6). With this approach, we were able to link species to their potential function in the soil food web and thus to their role in carbon and nitrogen cycling. A correlation network approach was used to visualize the strong potential interactions between all individual members of the soil food web. Only the positive correlations between species groups of Spearman's rank ≥ 0.9 were visualized. Within-group correlations were calculated but not displayed. To demonstrate that the pattern was robust, we also have displayed the co-occurrence matrix (Supplementary Figs 1 and 2).

**Soil properties.** Analyses of soil properties were performed by the Laboratoire d'analyse des sols d'Arras de INRA (Lille, France, http://www.lille.inra.fr/las). Soil samples were randomized before physicochemical characterizations in order to avoid any batch effects. The cation exchange capacity was determined by extraction with $Co(NH_3)_6Cl_3$ (ref. 56). Soil pH was measured on a soil slurry (1:5 deionised water:soil) following the ISO 10390 standard procedure. Total carbon (C), total nitrogen (N) and organic matter contents were measured after combustion at 1,000 °C (refs 57,58). Phosphorus (P) content was determined by $NaHCO_3$ (0.5 M, pH:8.5) extraction[59,60]. Exchangeable cations (Ca, Mg, Na, K, Fe, Mn and Al) were extracted using cobaltihexamine and determined by inductively coupled plasma spectrometry–atomic emission spectrometry. The most explaining soil properties for each of the groups are displayed in Supplementary Table 2. Fields where samples were collected from and the three sample sites within field are projected on the soil properties in Supplementary Fig. 9.

**Statistics on networks and communities of biota.** We analysed the number of species per aggregated group (Supplementary Fig. 3 and Supplementary Table 1) in three ways: the effect of site, succession, and time since abandonment. The sites OR, REY and TW (Supplementary Table 5) were categorized as recently abandoned fields; MO, NR and WV as mid-term abandoned fields; and MV, DK and BB as long-term abandoned fields. These categories mark the factor 'succession'. We also analysed the effect as a regression taking 'time since abandonment' as a continuous variable (Supplementary Table 1). For the other factors, we used a nested analysis of variance approach: when testing 'site' as a factor, subplots were nested in 'site' and when testing 'succession' as a factor, sites were nested in 'succession'. Spearman's rank correlation matrix was performed using R[61]. The principal component analysis/detrented correspondence analysis, canonical correspondence analysis/redundancy analysis on soil properties and non-metric dimensional analysis/principal coordinate analysis on soil community assemblage (Table 2 and Supplementary Table 2) were performed using CANOCO 5 (ref. 62). The analysis of similarities on the variation between and within successional stages owing to soil properties was performed in PAST3.X (ref. 63) (Supplementary Fig. 4) as well as the permutational multivariate analysis of variance and analysis of similarities in Table 2.

**Collecting the soil cores.** Between 23 July and 3 August 2012, we collected 90 intact soil cores from the same sampling points visited in 2011 (Supplementary Fig. 7, Supplementary Table 5). There were nine sites and three subsites. We collected three cores from each subsite for the three time points after labelling. This makes $9 \times 3 \times 3 = 81$ soil cores. From each site, we collected an additional soil core serving as a non-labelled control, which results in 90 intact soil cores in total that were collected from the field. Soil cores were collected 1 week before labelling to allow the microbial and faunal communities to stabilize after collecting and transportation. Cores were made using a 12-cm diameter soil corer. All cores were 20 cm deep. Immediately after collection, the intact cores were slid into a polyvinylchloride tube and closed with a fitting cap underneath to prevent respiration from the exposed soil. All cores were collected within 2 weeks under similar weather conditions.

**Labelling of the soil cores.** To complement the network analysis and to determine the effects of time since abandonment on carbon and nitrogen cycling in the soil, stable isotope probing of the intact cores was performed using dual labelled $^{15}$N ammonium nitrate ($^{15}NH_4$ $^{15}NO_3$) and $^{13}$C supplied to the plants in the form of $^{13}CO_2$ (ref. 64). The food web structure was resolved by identifying the microbes using phospholipid markers and identifying soil fauna morphologically combined with isotopic measurements.

One week prior to labelling with $^{13}$C, 81 intact soil cores with native vegetation were labelled with 10 atom% $^{15}$N ammonium nitrate ($^{15}NH_4^{15}NO_3$) (Sigma Aldrich). The amount of ammonium nitrate added was 0.1 mg per core, which corresponds to approximately 0.025 mg kg$^{-1}$ soil. The labelled ammonium nitrate solution was watered on the soil surface. As the potential N mineralization in all the soils was >5 mg kg$^{-1}$ week$^{-1}$, this was assumed not to disturb the system. The nine control cores were treated with the same amount of unlabelled ($^{14}$N) ammonium nitrate. The 81 cores were labelled with 99.99 atom% $^{13}CO_2$ (Cambridge Isotope Laboratories, Andover, MA, USA) in an artificially lit air-tight growth chamber for a total of 13 h. We placed nine cores, one from each

field, in the same chamber and kept under identical conditions but with a $^{12}CO_2$ atmosphere, representing the control treatment. The $CO_2$ concentrations in the chambers were monitored throughout the experiment. Prior to the start of labelling, the plants were allowed to assimilate carbon until the $CO_2$ concentration fell to 300 p.p.m. During this period, the photosynthetic rate was determined. When the $CO_2$ concentration of 300 p.p.m. was reached, $^{13}CO_2$ was injected into the chamber using a gas tight pumping system until the $CO_2$ concentration reached 450 p.p.m. During the labelling period, additional $^{13}CO_2$ was injected when the concentration fell below 350 p.p.m. In total, about 4.5 l of $^{13}CO_2$ was injected into the chamber. The plants were labelled during 8 h in the light, interrupted by 6 h of non-labelling in the dark during which no $^{13}CO_2$ was added and excess $CO_2$ was removed.

After labelling and the dark period, all cores were removed from the chambers and samples were collected from cores from both the $^{13}CO_2$ and $^{12}CO_2$ treatment after 1 day (27 cores, three per field), 1 week (27 cores) and 2 weeks (27 cores) after pulse labelling. The sampling strategy is presented in Supplementary Fig. 10. In short, samples of fauna, nematodes, enchytraeids and microbes (PLFA/NLFA) were collected. Earthworms and larger soil fauna, such as beetles, if present, were collected separately and stored in ethanol. Subsamples of soil were used to determine soil moisture and nutrient contents and to analyse the soil isotopic composition. Plant material was divided into root and shoot fractions, weighed, freeze dried and analysed for isotopic signatures. Roots were washed and air dried prior to the analyses. A subset of the root material was used for the nematode extractions. Part of the root and shoot material and soil was immediately frozen and freeze dried prior to the analyses of isotopes and extraction of PLFAs. The different groups of microbes, consumers and predators were displayed at the time point where most label was incorporated, microbes at 1 day, consumers at 1 week and predators at 2 weeks after labelling[65].

**$^{13}C$ and $^{15}N$ in the different parts of plant and soil biota.** Freeze-dried plant parts (shoots and roots) were ground to mesh size 0.1 μm. The $\delta^{13}C$ and $\delta^{15}N$ values of the samples were determined using an elemental analyser (Flash2000, Thermo) coupled to an isotope ratio mass spectrometer (IRMS, Thermo) to determine the amount of photosynthates allocated to and nitrogen assimilated by aboveground and belowground parts. Similarly, freeze-dried soil was used to determine the isotopic signatures in soil. Earthworms and handpicked spiders were freeze-dried and ground prior to the analysis of isotopic signatures. Enchytraeids and nematodes were individually picked from their liquid solutions under a microscope using a pig hair glued to a wooden stick. They were transferred into a tin capsule with a droplet of water and left to dry overnight before the tin capsules were closed. Nematodes were separated into root-feeders, fungivores, bacterivores and omni-carnivores by their mouth parts. Dependent on their individual weight, we needed around 60–100 individuals of root-feeding nematodes to reach the detection limit for IRMS. Micro-fauna was transferred into a tin capsule with a similar procedure using forceps and brushes. We separated all extracted micro-fauna into herbivorous (feeding on shoot material) cryptostigmatic and prostigmatic mites and herbivorous varia (others), fungivorous cryptostigmatic, astigmatic and prostigmatic mites and fungivorous collembola. We also separated predaceous mesostigmatic and prostigmatic mites and predaceous varia (small spiders). For each core, these 10 different groups were individually weighed and placed into separate tin capsules.

The incorporation of $^{13}C$ and $^{15}N$ into plants and soil was expressed as the increase of atom% $^{13}C$ and atom% $^{15}N$ values relative to the atom% $^{13}C$ and atom% $^{15}N$ values of unlabelled control plants and soil (excess atom% $^{13}C$ and excess atom% $^{15}N$). $\delta^{13}C$ and $\delta^{15}N$ values were calculated using the following formulas described by Werner and Brand[66]: $\delta^{13}C = (^{13}C/^{12}C_{sample}/^{13}C/^{12}C_{VPDB} - 1) \times 1000$ and $\delta^{15}N = (^{15}N/^{14}N_{sample}/^{15}N/^{14}N_{air-N2} - 1) \times 1000$. VPDB and Air-$N_2$ was used as reference values in equations. For further calibration, a standard curve was created using USGS40 ($\delta^{13}C$: − 26.39, $\delta^{15}N$: − 4.52), USGS41 ($\delta^{13}C$: + 37.63, $\delta^{15}N$: + 47.57), NIST8542 ($\delta^{13}C$: − 10.45) and USGS25 ($\delta^{15}N$: − 30.41) to which samples were corrected[67]. Atom% were calculated using the following equation: atom% $^{13}C = (^{13}C/^{12}C + ^{13}C) \times 100$ and atom% $^{15}N = (^{15}N/^{14}N + ^{15}N) \times 100$. Atom% excess $^{13}C$ and atom% excess $^{15}N$ were calculated by subtracting the atom% of unlabelled controls from the enriched samples.

Subsequently, carbon and nitrogen contents (unit) were calculated using the TCD trace of the EA analyser using a linear standard curve of different amounts of sulfanilamide (41.84% C, 16.27% N, Thermo), nicotinamide (59.01% C, 22.94% N, Thermo) and L-aspartic acid (36.09% C, 10.52% N, Thermo).

**Analyses of PLFAs and NLFAs.** PLFAs and NLFAs were extracted from the freeze-dried soil according to Boschker[68] and concentrations and $\delta^{13}C$ values were measured on a Thermo Trace Ultra gas chromatograph coupled to a Thermo Scientific Combustion Interface III and a Thermo Scientific Delta V IRMS. The internal standard methyl nonadecanoate fatty acid (19:0) was used for calculating concentrations. Three C20:0 methyl esters (Schimmelmann, Biogeochemical Laboratories, Indiana University) were used for calibration of the delta value. Identification of the compound was based on a BAME mix (Supelco 47080- u) and a FAME mix (Supelco 18919-1AMP). The following fatty acids were used as biomarkers for bacterial biomass: i14:0, i15:0, a15:0, i16:0, 16:1ω7t, 17:1ω7,

a17:1ω7, i17:0, cy17:0, 18:1ω7c, and cy19:0 (ref. 69). PLFA10Me16:0 was used as specific indicator for actinomycetes[70]. PLFA 18:2ω6.9 was considered as an indicator for fungal biomass[71,72]. The NLFA marker 16:1ω5 was used as an indicator of AMF[73,74]. The percentage of $^{13}C$ allocated to a certain PLFA was calculated from the amount of $^{13}C$ in each PLFA and total $^{13}C$ accumulation (excess $^{13}C$ pmol g$^{-1}$) in all PLFAs used as biomarkers for different microbial groups and these values were used in data analyses.

**Statistics on labelling data.** We analysed the effect of land abandonment as follows: the sites OR, REY and TW (Supplementary Table 1) were categorized as recently abandoned fields; MO, NR and WV as mid-term abandoned fields; and MV, DK and BB as long-term abandoned fields. These categories mark the factor 'succession' and were analysed with a Generalized Linear Model with nested design. Site was nested in 'succession', and the excess data were square-root transformed to meet the normality assumption. The analyses for $^{13}C$ excess data as well as for $^{15}N$ excess data were carried out in the same way. Analyses were performed in STATISTICA[75].

**Data availability.** The sequencing data are stored in Sequence Read Archive and can be found under accession numbers SRP049204 and SRP044011. All other data are available in the NIOO repository via http://mda.vliz.be/mda/directlink.php?fid=VLIZ_00000444_583ea8cd3f60c.

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

## Acknowledgements

This study was carried out as part of the EcoFINDERS research project (EU-FP7-264465), NWO-ALW-Veni 863.15.021 to E.M. and ERC-Adv 260-55290-SPECIALS to W.H.v.d.P. We thank Rebecca Pas for processing labelled material; Eefje Sanders for extracting PLFAs and NLFAs and sorting enchytraeids, earthworms and spiders for the labelling experiment; Thomas Verschut for sorting labelled mites and

collembolans; Javier Moliner Urdiales and Bekir Faydaci for collecting living soil-cores for the pulse labelling experiment; Agata Pijl, Valeria Bianciotto, Claudia Bragalini, Marine Peyret-Guzzon, Herbert Stockinger and Diederik van Tuinen for preparing libraries for pyrosequencing; and Peter de Vries for drawing Supplementary Fig. 7. The UMR1136 is supported by the ANR through the Laboratory of Excellence Arbre (ANR-11-LABX-0002-01). We thank George Kowalchuk, Peter de Ruiter, Thomas Crowther and Kelly Ramirez for their valuable comments on the manuscript. This is NIOO-KNAW publication 6199.

## Author contributions

W.H.v.d.P. and P.L. were involved in designing the field survey; E.M., P.P., W.D., J.H.F. and W.H.v.d.P. collected the soil samples for the network analyses. E.M. and S.E.H. designed and performed the pulse labelling experiment, W.d.B. and J.A.v.V. advised about the pulse labelling experiment, N.R.H. performed the EA-IRMS analyses for the labelling experiment, H.Z. performed the GC-c-IRMS analyses for the pulse labelling experiment; L.B.S. performed the network analyses, M.-L.B., M.G., B.C.T., R.I.G., D.R., M.B., S.U., F.M., P.P. and S.E.H. were involved in the pyrosequencing of AMF, bacteria, archaea and fungi; M.B., E.T. and M.d.H. were involved in the bioinformatics; S.G., M.B. and A.W. identified and analysed protozoa, H.D. identified and R.L.S. sorted labelled nematodes, H.-B.J. analysed PLFAs, R.M.S. identified enchytraeids, W.D. mites, J.J. collembolans, O.S. earthworms and P.P. provided the environmental data. E.M., S.H.E. and L.B.S. did the data analysis and statistics; E.M., S.E.H. and W.H.v.d.P. wrote the manuscript; all others co-commented on the manuscript.

## Additional information

**Competing financial interests:** The authors declare no competing financial interests.

**Publisher's note**: 

