## [Peer Review File · Nature Communications]

Editorial Note: This manuscript has been previously reviewed at another journal that is not operating a transparent peer review scheme. This document only contains reviewer comments and rebuttal letters for versions considered at Nature Communications. Mentions of prior referee reports have been redacted.

PEER REVIEW FILE

Reviewers' Comments:

Reviewer #1 (Remarks to the Author):

Morrien et al. have ably addressed all of the reviewers' remarks. I still believe that there are limitations to their study. In particular, the use of atom% excess values as opposed to mass ^{13}C label data makes their data less compelling. For example, atom% ^{13}C excess in roots is not independent of biomass. I would love to know how much ^{13}C label was actually transported below ground to the roots, but their approach does not permit this and without root biomass data we do not know if those atom% excess values actually translate to different C uptake amounts by the soil food web. Simply putting this caveat into the Methods, however, would appease me so that the general reader can assess the work in full knowledge of all the limitations. I suggest this because, taking a step back, if you'd found no effects on C movement, then I would have said that your methods were inappropriate. But given the controlled nature of the work, and the fact you do find marked differences among land uses in ^{13}C excess values moving through the food web, suggests to me that the approach did yield real insights. With all the additional analyses (in particular the permutations), I also feel that the network conclusions are robust and appropriately discussed in terms of correlations/ associations (because ecological causation is not yet clear in any network analyses).

Together then, I think your overall data are sufficient to support your claim that there is a transition toward fungi moving more of the carbon and with increasingly efficient flows through the foodweb, in advance or in place of any shift toward changing bacterial to fungal ratios. The results then are a major finding because they suggest the need to rethink how we consider relationships between soil foodweb structure and carbon cycling in soils.

Two other (very) minor concerns. One lines 197 and 207 you still state that it was not possible to analyze ^{15}N in microbes. Maybe this is a language nuance that did not come across in my first review. Yes, it is possible to analyze ^{15}N in microbes and I explained how. You chose not to

analyze it. That is very different. The paper goes to a general readership and so it should be clear it's possible but you chose not to. As such, the wording should be along the lines of, "As we chose not to analyze ^{15}N in microbes..."

Second, you give a lot of extended data but often not what we'd want to see. Why give $\delta^{13}\text{C}$ ratios for example? You don't use them and in a labeling study of this magnitude would be wrong to (given nonlinearity between δ and atom% at highly enriched values). Please give instead the atom% ^{13}C excess values. Overall, it just shows greater awareness of the stable isotope tracer approaches employed.

Reviewer #4 (Remarks to the Author):

This is a transferred manuscript from Nature to NComms. In the first submission (Nature) I was referee #4 and was now asked to review how well the authors responded to my comments and to the comments of ref #2 and #3.

Generally, the changes have improved this paper as the writing and meaning of terms is clearer in the revised version. However, the changes mostly comprise wording of individual sentences, while the authors often do not agree with the comments and have repudiated substantial changes of their manuscript. However, as it is a new submission to another journal this is partly understandable.

Nonetheless, I am not satisfied how my concerns have been accounted for. I am still thinking that presenting some more data e.g. on plant diversity, root biomass and litter quality (my third comment) and discussing how these data are related to the findings would improve the manuscript. Referencing in the response letter to papers where these requested data have been published does not help to increase the quality of the paper as future reader might be interested, too. Thus, please indicate in the main text where these data can be found and how they are related to your data.

My 4th comment (shift in the microbial composition in the rhizosphere) was not intended to see even more data; I wanted the authors to give a more mechanistic explanation of this study. So far this study just says "tightening of the belowground network increases C uptake". But the authors failed to explain it. Stating that answering my comment is "outside the scope of this article" is really a weak response! It would increase the quality of the article if the authors could demonstrate a mechanism or at least try to discuss mechanisms underlying the higher C uptake in the fungal channel when the network is tighter. Possible explanations are community shifts in the rhizosphere, higher fungal turn-over due to higher predation on fungi, higher AMF colonization of plants in later sessional stages, etc.

L66-68: It is weird that the implication of this study is that a proper management is needed for

efficient carbon storage, as 1) this study was set up on abandoned / unmanaged land with highest effects long time after the management has stopped (what would be a management recommendation by the authors?). 2) What do the authors mean with efficient carbon storage? Carbon that is stored as organic matter in soil, carbon that is stored in the soil microbial community and in animals? The latter is considered as short term, compared to organic C in soils (see comments of ref #3). The study found higher C uptake by several taxa of the fungal channel (without an increase of fungal biomass). The higher uptake indicates a faster C transfer through the soil network that in turn indicates a higher/faster mineralisation of organic plant derived carbon. This finally implies less carbon storage in soil when plant C input has not increased (root data!). Please delete or rewrite the last sentence of the abstract AND consider this argumentation when discussing the results.

L225-226: This ratio alone does not give any information, particular as it does not represent the microbial biomass. Please convert the ratio of fungal-to-bacterial markers into microbial C (see my comment from the previous review).

Response to referee #2:

Most of the points made by referee #2 have been adequately accounted for. But the critique of the too low number of replicates is still valid and could of course not be changed. However, in my opinion the authors should have discussed this shortcoming of their manuscript with the implication for generalization of the results. Also the biggest concern of the ref #3 is in my opinion not sufficiently answered. By contrasting the variation of the similarities between groups and within groups (response to 3rd comment of ref #2) does not give a direct statistical answer about the role of abiotic soil properties (the authors refer in their response to extended data table 10, but it is not attached to the revised version). Here a stat test is needed testing the direct effect of homogenised soil properties over time and tightening network.

Moreover, the C uptake of most fungal feeding groups is highest in mid-term abandoned fields and similarly low in short and long term abandoned fields (Extended Data Table 8, and see below), how does this correspond with stronger network tightening in long-term abandoned than in short term. Please discuss!

Response to referee #3:

Also, most points made by referee #3 have been accounted for. However, ref#3 was not convinced the presented results support an increased efficiency in C sequestration /uptake into different trophic levels (main point 3 and specific comments). In their response to point 3 the authors present statistics on the uptake of ^{13}C by different taxa (Extended Data Table 8). I found the result not very convincing and supportive to their statement in the abstract that during the course of succession "...higher trophic levels in the fungal channel took up more carbon". Only three out of five of the considered groups feeding on fungi differ significantly in their uptake among the successional stages. From these tree taxa cryptostigmatic mites shows increasing

trend over time, while uptake of prostigmatic mites and astigmatic mites do not differ in the early and late stage succession, the mid-succ stage differs.

Reviewers' comments:

Reviewer #1 (Remarks to the Author):

Morrien et al. have ably addressed all of the reviewers' remarks. I still believe that there are limitations to their study. In particular, the use of atom% excess values as opposed to mass 13C label data makes their data less compelling. For example, atom% 13C excess in roots is not independent of biomass. I would love to know how much 13C label was actually transported below ground to the roots, but their approach does not permit this and without root biomass data we do not know if those atom% excess values actually translate to different C uptake amounts by the soil food web. Simply putting this caveat into the Methods, however, would appease me so that the general reader can assess the work in full knowledge of all the limitations. I suggest this because, taking a step back, if you'd found no effects on C movement, then I would have said that your methods were inappropriate. But given the controlled nature of the work, and the fact you do find marked differences among land uses in 13C excess values moving through the food web, suggests to me that the approach did yield real insights. With all the additional analyses (in particular the permutations), I also feel that the network conclusions are robust and appropriately discussed in terms of correlations/ associations (because ecological causation is not yet clear in any network analyses).

→ We thank the reviewer for constructive comments: indeed our analysis of 13C incorporation has its limitations, but still we see marked differences in 13C excess values moving through the food web. As proposed by the referee, we have mentioned this as a caveat of our approach in the Methods section. It now reads as:

*"The incorporation of ^{13}C and ^{15}N into plants and soil was expressed as the increase of atom% ^{13}C and atom% ^{15}N values relative to the atom% ^{13}C and atom% ^{15}N values of unlabelled control plants and soil (excess atom% ^{13}C and excess atom% ^{15}N). $\delta^{13}\text{C}$ and $\delta^{15}\text{N}$ values were calculated using following formulas described by Werner & Brand⁷³: $\delta^{13}\text{C} = [^{13}\text{C}/^{12}\text{C}_{\text{sample}} / ^{13}\text{C}/^{12}\text{C}_{\text{VPDB}} - 1] * 1000$ and $\delta^{15}\text{N} = [^{15}\text{N}/^{14}\text{N}_{\text{sample}} / ^{15}\text{N}/^{14}\text{N}_{\text{air-N2}} - 1] * 1000$. VPDB and Air-N2 was used as reference values in equations. For further calibration a standard curve was created using USGS40 ($\delta^{13}\text{C}$: -26.39, $\delta^{15}\text{N}$: -4.52), USGS41 ($\delta^{13}\text{C}$: +37.63, $\delta^{15}\text{N}$: +47.57), NIST8542 ($\delta^{13}\text{C}$: -10.45) and USGS25 ($\delta^{15}\text{N}$: -30.41) to which samples were corrected⁷⁴. Atom% were calculated using the following equation: atom% $^{13}\text{C} = (^{13}\text{C} / ^{12}\text{C} + ^{13}\text{C}) * 100$ and atom% $^{15}\text{N} = (^{15}\text{N} / ^{14}\text{N} + ^{15}\text{N}) * 100$. Atom% excess ^{13}C and atom% excess ^{15}N were calculated by subtracting the atom% of unlabelled controls from the enriched samples. Subsequently, carbon and nitrogen contents (unit) were calculated using the TCD trace of the EA analyser using a linear standard curve of different amounts of Sulfanilamide (41.84 %C, 16.27% N, Thermo), Nicotinamide (59.01 %C, 22.94% N, Thermo) and L-aspartic acid (36.09 %C, 10.52% N, Thermo)."*

→ In addition, we have separated Figure 7 (total biomass) into a figure presenting root and shoot mass (as is also proposed by Referee 4).

Together then, I think your overall data are sufficient to support your claim that there is a transition toward fungi moving more of the carbon and with increasingly efficient flows through the food web, in advance or in place of any shift toward changing bacterial to fungal ratios. The results then are a major finding because they suggest the need to rethink how we consider relationships between soil food web structure and carbon cycling in soils.

→ We agree.

Two other (very) minor concerns. One lines 197 and 207 you still state that it was not possible to analyze 15N in microbes. Maybe this is a language nuance that did not come across in my first

review. Yes, it is possible to analyze ^{15}N in microbes and I explained how. You chose not to analyze it. That is very different. The paper goes to a general readership and so it should be clear it's possible but you chose not to. As such, the wording should be along the lines of, "As we chose not to analyze ^{15}N in microbes..."

→ Agree that ^{15}N can be analyzed in microbes as a whole group. Indeed, we would have been interested in discriminating between ^{15}N in bacteria and fungi and arbuscular mycorrhizal fungi, but that was not possible by chloroform fumigation (only the ^{15}N label in the total microbial biomass could be determined; the reviewer agrees with us based on the formulation in the review of our first submission). As we could not distinguish between ^{15}N in bacteria and fungi, we chose not to present ^{15}N in microbes at all and not to relate tightening of the belowground networks to the microbial efficiency of transferring nitrogen. We have justified this choice in the main text by stating that: *It is possible to analyse ^{15}N in microbes, but methods do not allow data to be separated into bacterial and fungal ^{15}N . Therefore, we chose not to relate tightening of the belowground networks to the microbial efficiency of... Etc.*

Second, you give a lot of Supplementary but often not what we'd want to see. Why give $\delta^{13}\text{C}$ ratios for example? You don't use them and in a labeling study of this magnitude would be wrong to (given nonlinearity between δ and atom% at highly enriched values). Please give instead the atom% ^{13}C excess values. Overall, it just shows greater awareness of the stable isotope tracer approaches employed.

→ We see the point of the referee, but during the first review round, it has been requested to provide $\delta^{13}\text{C}$ values of enriched samples versus non-enriched samples. These raw data can be found in Supplementary table 9. In Supplementary Table 8 we have now converted the excess ^{13}C label data into atom%, however, this did not affect the outcome nor the conclusions as these values are linearly correlated ($R=0.9999$). We initially used the δ values as we have a broad range of values (i.e. soils close to natural abundances while plant material was highly labeled.). We agree that using atom% excess values for these type of data is better and therefore we changed them in Supplementary Table 8. However, we decided to keep the ^{13}C excess data presented in Figure 2 because if we would convert these data into atom% it would no longer be possible to show the data for bacteria, fungi and AMF in the same figure as these groups are expressed in ^{13}C excess in pmol per gram soil. We cannot convert these data into atom% and although this is not strictly similar as ^{13}C excess data, it falls within the same range, while atom% are not.

Reviewer #4 (Remarks to the Author):

This is a transferred manuscript from Nature to NComms. In the first submission (Nature) I was referee #4 and was now asked to review how well the authors responded to my comments and to the comments of ref #2 and #3.

Generally, the changes have improved this paper as the writing and meaning of terms is clearer in the revised version. However, the changes mostly comprise wording of individual sentences, while the authors often do not agree with the comments and have repudiated substantial changes of their manuscript. However, as it is a new submission to another journal this is partly understandable.

→ We thank the referee for re-reviewing the manuscript and it is a pity that our substantial revision and the rebuttal suggested that we were not willing to make substantial changes. We had quite some suggestions to focus on and perhaps some issues were wrongly estimated to be less important than others. Therefore, it is nice that we have another opportunity to address these remaining issues as

well.

Nonetheless, I am not satisfied how my concerns have been accounted for. I am still thinking that presenting some more data e.g. on plant diversity, root biomass and litter quality (my third comment) and discussing how these data are related to the findings would improve the manuscript. Referencing in the response letter to papers where these requested data have been published does not help to increase the quality of the paper as future readers might be interested, too. Thus, please indicate in the main text where these data can be found and how they are related to your data.

→ We have added data on plant diversity by including the H index in Supplementary Figure 5, and adding root (and shoot) biomass by subdividing the total biomass in Supplementary Figure 7 into roots and shoots. We do not have information on litter quality, but we have added C/N values of the shoots/roots in Supplementary Figure 7 as to give a proxy of quality differences that might be found back in the litter data.

My 4th comment (shift in the microbial composition in the rhizosphere) was not intended to see even more data; I wanted the authors to give a more mechanistic explanation of this study. So far this study just says “tightening of the belowground network increases C uptake”. But the authors failed to explain it. Stating that answering my comment is “outside the scope of this article” is really a weak response! It would increase the quality of the article if the authors could demonstrate a mechanism or at least try to discuss mechanisms underlying the higher C uptake in the fungal channel when the network is tighter. Possible explanations are community shifts in the rhizosphere, higher fungal turn-over due to higher predation on fungi, higher AMF colonization of plants in later successional stages, etc.

→ Sorry about this misunderstanding, but some of the other reviewers wanted to see more data, which explains that aspect, whereas we did not want to be too speculative in the discussion. That is why we worded our rebuttal in terms of ‘outside the scope of this study’. Nevertheless, the comments of the reviewer and the Associate Editor on possible mechanisms of network tightening stimulated to include extra paragraphs where we discuss possible mechanisms of why the networks increase in density, and why this ‘tightening’ could lead to higher C uptake in the fungal channel. This is what we added about tightening and higher C uptake in the fungal channel:

The novel combination of correlation-based network analysis and isotope labelling shows that during land abandonment soil networks become more tight and that efficiency of the carbon uptake in the fungal channel of the soil food web increases, without an increase in the total amount of soil biodiversity, or in fungal to bacterial biomass ratios. For nitrogen, the non-microbial species groups revealed a similar pattern as for carbon. Tightening of the networks reflects stronger co-occurring patterns of variation in soil biota²⁶. Increased carbon and nitrogen uptake capacity by the fungal channel in the soil food web can be explained by stronger co-occurrence of preys and their predators²⁵, which enhances the efficiency of resource transfer in the soil food web compared to a soil food web where preys and predators are spatially isolated.

Increased network tightening may be due to several factors. First, tightening may be caused by successional shifts in species²⁷. Bacteria and fungi showed hump-shaped development in numbers of taxa, whereas numbers of AMF taxa steadily increased, indirectly suggesting that there are indeed shifts in species composition along the successional gradient. AMF have been suggested to increasingly influence plant community composition with increasing time since land abandonment²⁸. However, in our study network tightening is due to changes in more species groups than AMF alone. Second, increased tightening could be due to declined nutrient availability in the soil along the successional gradient^{19,29-30}, which may enhance the necessity of stronger cooperative and trophic interactions between functional groups of soil biota.

Third, changes in the soil physical conditions can influence network tightening³¹. Arable soils are assumed to be relatively heterogeneous^{32,33}, whereas natural succession following land abandonment will increase spatial heterogeneity in abiotic soil conditions³⁴. Soil biota have a variety of responses to soil heterogeneity³⁵. Increased soil heterogeneity could contribute to network tightening, when it enhances co-occurrence patterns of variation in the soil biota. We found reduced variation in soil abiotic properties from recent to mid/term abandoned fields, but there were no differences in variation between recent and longer-term abandonment stages, which only partly supports the possibility that changes in soil abiotic factors enhance network tightening. Further correlative analyses of soil abiotic properties and network tightening would require independent pairs, however, we do not have individual networks for each individual soil sample used for abiotic analyses.

L66-68: It is weird that the implication of this study is that a proper management is needed for efficient carbon storage, as 1) this study was set up on abandoned / unmanaged land with highest effects long time after the management has stopped (what would be a management recommendation by the authors?). 2) What do the authors mean with efficient carbon storage? Carbon that is stored as organic matter in soil, carbon that is stored in the soil microbial community and in animals? The latter is considered as short term, compared to organic C in soils (see comments of ref #3). The study found higher C uptake by several taxa of the fungal channel (without an increase of fungal biomass). The higher uptake indicates a faster C transfer through the soil network that in turn indicates a higher/faster mineralisation of organic plant derived carbon. This finally implies less carbon storage in soil when plant C input has not increased (root data!). Please delete or rewrite the last sentence of the abstract AND consider this argumentation when discussing the results.

→We agree that this has been confusing and as referee 1 has phrased it: *that there is a transition toward fungi moving more of the carbon and with increasingly efficient flows through the food web, in advance or in place of any shift toward changing bacterial to fungal ratios. The results then are a major finding because they suggest the need to rethink how we consider relationships between soil food web structure and carbon cycling in soils.*

We have used this phrasing to reformulate the text fragments indicated above by reviewer 4 (also avoiding discussion about management, for example whether abandonment is a management strategy or not).

The end of the Abstract now reads as:

'The implication of our findings is that during nature restoration the efficiency of nutrient cycling and carbon uptake can increase by a shift in fungal composition and/or fungal activity without an increase in fungal to bacterial biomass ratio. Therefore, we propose that relationships between soil food web structure and carbon cycling in soils need to be reconsidered.'

The end of the Discussion reads as:

'Our results suggest that transition in fungal composition can change element cycling and carbon uptake in soil without an increase in fungal biomass or fungal to bacterial biomass ratio. We propose that there is a need to verify these findings also in other chronosequences, and re-think how soil food web structure influences carbon cycling in soils.'

L225-226: This ratio alone does not give any information, particular as it does not represent the microbial biomass. Please convert the ratio of fungal-to-bacterial markers into microbial C (see my comment from the previous review).

→ We used the reference from the previous review round to convert the fungi and bacterial markers into microbial C. But as it uses estimates to convert we strongly feel uncomfortable about this conversion factor as it generalizes all bacteria and all fungi to have the same tissue properties, which is most likely not a reflection of reality. Moreover, we feel it is more insightful to stay closer to measured values than to convert using estimates. Also, when we check recent other studies using C labelled data in microbes, none of them uses this conversion factor. The information that we want to get across here is the ratio between fungi and bacteria.

Response to referee #2:

Most of the points made by referee #2 have been adequately accounted for. But the critique of the too low number of replicates is still valid and could of course not be changed. However, in my opinion the authors should have discussed this shortcoming of their manuscript with the implication for generalization of the results.

→ This point has several elements: First there is the number of replicates, then there is the fact that we have used one chronosequence. Even when we would have increased the number of replicates, we would still have one chronosequence. The number of replicates will have made our results less sensitive for picking up statistical significant effects. Therefore, our study provides a conservative assessment. As we do not have data on other chronosequences (this was already an immense effort), we have added that point to the discussion: *'We propose that there is a need to verify these findings also in other chronosequences, and re-think how soil food web structure influences carbon cycling in soils.'*

Also the biggest concern of the ref #3 is in my opinion not sufficiently answered. By contrasting the variation of the similarities between groups and within groups (response to 3rd comment of ref #2) does not give a direct statistical answer about the role of abiotic soil properties (the authors refer in their response to Supplementary Table 10, but it is not attached to the revised version). Here a stat test is needed testing the direct effect of homogenised soil properties over time and tightening network.

→ Contrasting the variation of the similarities between groups and within groups: sorry for referring to Supplementary Table 10 in the rebuttal; we did it correct in the revised manuscript, where we referred to Supplementary **Figure 10**. About the statistical test that the referee suggests: This is of course what should be done. However, we only have one network per succession stage (Supplementary Table 6). Therefore, we used the variation in soil properties compared to the similarities of the groups on which the networks are based. We think that this is not wrong at all and the best that we could do in the given circumstances. In conclusion, our data do not allow to produce such a correlation analysis. We have discussed this in the revised manuscript.

Moreover, the C uptake of most fungal feeding groups is highest in mid-term abandoned fields and similarly low in short and long term abandoned fields (Supplementary Table 8, and see below), how does this correspond with stronger network tightening in long-term abandoned than in short term. Please discuss!

→ We have discussed this in the new paragraph that explains network tightening (see our rebuttal to reviewer 4 above).

Response to referee #3:

Also, most points made by referee #3 have been accounted for. However, ref#3 was not convinced the presented results support an increased efficiency in C sequestration /uptake into different

trophic levels (main point 3 and specific comments). In their response to point 3 the authors present statistics on the uptake of ^{13}C by different taxa (Supplementary Table 8). I found the result not very convincing and supportive to their statement in the abstract that during the course of succession "...higher trophic levels in the fungal channel took up more carbon". Only three out of five of the considered groups feeding on fungi differ significantly in their uptake among the successional stages. From these tree taxa cryptostigmatic mites shows increasing trend over time, while uptake of prostigmatic mites and astigmatic mites do not differ in the early and late stage succession, the mid-succ stage differs.

→Indeed, the fungal channel becomes more important during succession, but that there is variation among successional stages in who plays the most important role. In the new discussion on what explains network tightening and how this works out into network functioning, we have given the example of microbes and AMF to explain that the trends differ among species groups. We have not further addressed the case mentioned by the referee on the mites, because it is clear that not all groups responded equally strong, but the overall trend is an increase. In order to further address that point, it would be important not only to know that the various mites not only change abundance, but that they also can become replaced by other species when succession progresses. That could explain why networks can become more efficient even though the total number of taxa does not change. We have addressed this point for microbes in the new paragraph on what explains network tightening and added discussion in the revised manuscript (see the insert at the start of this rebuttal in the response to the comments made by the Editor).

Reviewer's Comments:

Reviewer #1 (Remarks to the Author):

No further comments

Reviewer #4 (unable to re-review)